# Study of Acoustic Emission Signal Noise Attenuation Based on Unsupervised Skip Neural Network

**DOI:** 10.3390/s24186145

**Published:** 2024-09-23

**Authors:** Tuoya Wulan, Guodong Li, Yupeng Huo, Jiangjiang Yu, Ruiqi Wang, Zhongzheng Kou, Wen Yang

**Affiliations:** 1School of Transportation, Inner Mongolia University, Hohhot 010031, China; 311993472@imu.edu.cn (T.W.);; 2Inner Mongolia Engineering Research Center of Testing and Strengthening for Bridges, Hohhot 010020, China; 3School of Construction Engineering, Dalian University of Technology, Dalian 116024, China

**Keywords:** acoustic emission, deep learning, noise attenuation

## Abstract

Acoustic emission (AE) technology, as a non-destructive testing methodology, is extensively utilized to monitor various materials’ structural integrity. However, AE signals captured during experimental processes are often tainted with assorted noise factors that degrade the signal clarity and integrity, complicating precise analytical evaluations of the experimental outcomes. In response to these challenges, this paper introduces an unsupervised deep learning-based denoising model tailored for AE signals. It juxtaposes its efficacy against established methods, such as wavelet packet denoising, Hilbert transform denoising, and complete ensemble empirical mode decomposition with adaptive noise denoising. The results demonstrate that the unsupervised skip autoencoder model exhibits substantial potential in noise reduction, marking a significant advancement in AE signal processing. Subsequently, the paper focuses on applying this advanced denoising technique to AE signals collected during the tensile testing of steel fiber-reinforced concrete (SFRC), the tensile testing of steel, and flexural experiments of reinforced concrete beam, and it meticulously discusses the variations in the waveform and the spectrogram of the original signal and the signal after noise reduction. The results show that the model can also remove the noise of AE signals.

## 1. Introduction

As a globally utilized construction material, concrete is essential for both edifices’ structural integrity and durability [1]. However, its limited ability to withstand high tensile loads and its propensity to become brittle often result in crack formation, presenting significant drawbacks in engineering applications [2]. Hence, steel fiber reinforced concrete (SFRC) has emerged as a robust enhancement material. By incorporating short steel fibers into traditional concrete, SFRC markedly improves the tensile strength, flexural resilience, and impact resistance. This modified concrete addresses conventional concrete’s inherent brittleness and poor crack resistance, which can compromise the structural durability and safety [3]. The steel fibers, uniformly distributed within the matrix, effectively connect and inhibit crack propagation, thereby significantly enhancing the mechanical properties and toughness of the concrete. Particularly well-suited to structures that are exposed to high-stress situations and dynamic loading conditions, SFRC finds extensive application in bridge engineering, thereby bolstering these critical infrastructures’ structural integrity and resilience [4].

In civil engineering, scrutinizing and evaluating concrete structure damage poses intricate challenges. In various diagnostic approaches, acoustic emission (AE) technology is recognized as an increasingly preferred non-destructive testing method, noted for its exceptional sensitivity and ability to assess conditions in real time. This technique identifies micro-cracks and other defects in the concrete by recording the acoustic signals emitted under states of stress and structural failure [5]. Consequently, this approach establishes a robust foundation for continuous structural health monitoring and accurate damage evaluation, thereby enhancing the reliability of assessments related to structural integrity.

In practical applications, AE signals are often compromised by various noise interferences during transmission, including environmental noise, electromagnetic disturbances, and instrumental noise from measuring equipment [6]. Such noise markedly reduces the signals’ quality and analytic precision. Consequently, effectively eliminating noise components becomes a pivotal challenge in processing AE signals. Conventional signal processing techniques include wavelet packet analysis [7], the Hilbert–Huang transform (HHT) [8], and empirical mode decomposition (EMD) [9]. For instance, wavelet packet analysis effectively distinguishes between the signal and noise across varied frequency bands through a multi-scale analysis [10]. Donoso et al. observed that this method successfully extracts signals within specific frequency bands [11].

Nonetheless, under conditions of complex noise, the performance of wavelet packet analysis can be less than optimal, and it is susceptible to parameter selection and adjustment [12]. The Hilbert HHT represents a productive methodology for addressing non-linear and non-stationary signals. Huang et al. demonstrated its superiority in managing random and non-linear signals by integrating EMD with the Hilbert spectral analysis [13]. However, the practical deployment of this technology requires significant expertise and experience and is characterized by substantial computational complexity. Furthermore, Hussain et al. confirmed its potential in addressing non-linear and non-stationary signals in their study on applying HHT to diagnose mechanical faults [14]. The method of empirical mode decomposition makes the signal processing more versatile by adaptively decomposing signals. Wu and Huang noted that while EMD presents certain advantages in noise reduction and feature extraction, its performance remains constrained in high-noise environments [15]. Meanwhile, Dragomiretskiy et al. introduced the ensemble empirical mode decomposition (EEMD), which enhances the decomposition results by introducing white noise. Despite this advance, the methodology still presents computational challenges [16].

Although these traditional methods exhibit some effectiveness under specific conditions and contexts, they show limited robustness in multi-noise environments and are characterized by a lower level of automation. Consequently, these methodologies struggle to satisfy the stringent requirements of high precision and real-time signal analysis required by complex engineering applications.

In recent years, with the advancement of artificial intelligence and machine learning technologies, deep learning (DL) methodologies have shown significant potential in signal processing [17]. Deep learning autonomously learns features within signals through multiple layers of non-linear transformations, enabling a more accurate capture and extraction of pertinent information from signals than traditional methods. Furthermore, deep learning approaches offer increased automation, significantly reducing the reliance on manually set parameters and expert knowledge. They can adaptively handle complex and variable noise environments, enhancing the accuracy and robustness of signal processing.

For other signal processing, deep learning methods have achieved considerable success. For instance, in image processing, convolutional neural networks (CNNs) effectively extract and recognize image features through mechanisms of local receptive fields and shared weights. A seminal work by Krizhevsky et al. demonstrated in their AlexNet model employed deeper convolutional kernels to enhance the precision and efficiency of image recognition [18]. Furthermore, the introduction of deep residual networks (ResNet) by He et al. extended the depth of convolutional neural networks, significantly enhancing their performance in image recognition tasks [19].

In the domain of speech processing, recurrent neural networks (RNNs) and their variants, such as long short-term memory networks (LSTMs), have shown robust capabilities in managing temporal data, particularly in speech recognition and text generation. Hannun et al. introduced a significant speech recognition model that incorporated LSTM structures, significantly enhancing both the performance and robustness of speech recognition systems [20]. Similarly, Graves et al. implemented a recurrent neural network-based connectionist temporal classification (CTC) method in speech recognition, substantially improving the recognition accuracy [21]. In natural language processing, the transformer model, based on the attention mechanism, employs self-attention to analyze and understand large-scale textual data efficiently. Introduced by Vaswani et al., the transformer model, with its entirely attention-based architecture, has shown an exceptional performance in tasks like machine translation and text generation [22]. Furthermore, the BERT model (bidirectional encoder representations from transformers), proposed by Devlin et al. (2018), leverages a bidirectional encoder framework to enhance the performance of natural language processing tasks dramatically [23].

In addition, deep learning has enjoyed numerous successful deployments in signal denoising. By employing deep learning-based autoencoder technologies, which meld multiple non-linear modules, input data are transformed into higher-order, abstract representations. This complex process automatically captures the intricate intrinsic relationships among data features, facilitating robust denoising [24]. Autoencoders, a classic algorithm within deep learning, excel at isolating and extracting valuable signals from data saturated with noise [25,26,27]. Typically comprising two principal components—the encoder and the decoder—autoencoders function by the encoder learning to represent the features of valuable signals within the input data and mapping these to an abstract feature space. Subsequently, the decoder reconstructs these abstract features and projects them back into the original data space, effectively attenuating the noise [28]. Within seismic data processing, researchers such as Chen [29] have developed an unsupervised sparse autoencoder that is specifically tailored for diminishing random noise in stacked seismic datasets. This sparse autoencoder has demonstrated exceptional denoising capabilities, which is particularly evident in its deployment across certain seismic datasets.

Similarly, Saad and Chen [30] proposed a deep denoising autoencoder model that initially leverages synthetic data for pre-training, followed by unsupervised random noise attenuation in real-field data. Song et al. developed a deep convolutional autoencoder neural network for denoising purposes, which is structured into distinct encoding and decoding frameworks. The encoding framework, composed of convolutional and pooling layers, is adept at extracting features from seismic data while concurrently mitigating noise [31]. Huang et al. introduced a novel denoising methodology termed A-SK22, which is an encoder–decoder network structure based on a hybrid attention mechanism with symmetric skip connections [32].

Consequently, this manuscript delineates a deep learning-based approach to denoise AE waveform signals in concrete in order to leverage the autonomous feature extraction and robust fitting capabilities inherent in deep learning models. This approach seeks to enhance the noise reduction performance of AE signals. Ultimately, the enhancement provides more accurate and reliable data support for the health monitoring of concrete structures.

## 2. Denoising Models

### 2.1. Traditional Denoising Models

(1)Wavelet Packet Model (WPM)

Wavelet packet denoising is a sophisticated signal-processing technique that is used to analyze non-stationary signals through multi-level decomposition. This technique comprises three main steps: signal decomposition, thresholding, and reconstruction.

The process begins by decomposing the signal *f*(*t*) into wavelet coefficients using the wavelet transform. This transform adapts the mother wavelet function *ψ*(*t*) with scaling (*a*) and shifting (*b*) factors:(1)$$\psi_{a,b}(t)=\frac{1}{\sqrt{a}}\psi \left(\frac{t-b}{a}\right)$$

These coefficients are organized hierarchically through wavelet packet decomposition, distributing the signal information across frequency bands. This arrangement allows for a detailed analysis by dividing the signal’s energy in both the time and frequency dimensions. By choosing suitable wavelet basis functions and decomposition levels, the technique distinguishes valuable information from noise, enhancing the signal clarity.

Once the decomposition is complete, thresholding selectively eliminates or reduces the wavelet coefficients to achieve denoising. Smaller coefficients often represent noise, while larger ones contain crucial signal information. Standard techniques include soft and hard thresholding:

Soft Thresholding: This method sets the coefficients below the threshold to zero and linearly reduces those above it, providing smooth noise attenuation.
(2)$$di=sgn(ci)\cdot max(\left|ci\right|-\lambda ,0)$$

Hard Thresholding: This technique sets the coefficients below the threshold to zero without altering the others, effectively handling impulsive noise.
(3)$$di=\left\{\begin{array}{l} ci,\;if\left|ci\right|>\lambda \\ 0,\;otherwise \end{array}\right.$$

Adaptive thresholding methods dynamically adjust the thresholds for optimal noise reduction, enhancing the technique’s effectiveness across varying signal conditions.

The final step is reconstructing the signal by applying an inverse transformation to the processed wavelet coefficients:
(4)$$f(t)=\sum_{a,b}c_{a,b}\psi_{a,b}(t)$$

This reconstruction reassembles the coefficients to approximate the original signal, minimizing new noise or distortion.
(2)Hilbert Transform Model (HTM)

The Hilbert transform is an effective method for noise reduction, extending an actual signal across the frequency spectrum to extract phase information and enhance denoising. By applying the Hilbert transform, a real signal *x*(*t*) becomes an analytic signal. This is calculated by:
(5)$$\hat{x}(t)=\frac{1}{\pi }P.V.\int_{-\infty }^{\infty }\frac{x(\tau )}{t-\tau }d\tau$$

The analytic signal is:
(6)$$z(t)=x(t)+j\hat{x}(t)$$

Here, *j* is the imaginary unit, which helps to expand the signal’s spectrum and facilitates the extraction of the instantaneous amplitude and frequency, which is particularly beneficial for signals with narrow frequency components. Next, thresholding manipulates these instantaneous characteristics. The instantaneous amplitude *A*(*t*) and frequency *ω*(*t*) are calculated as:
(7)A(t)=|z(t)|
(8)$$\omega (t)=\frac{d\phi (t)}{dt}$$
where *ϕ*(*t*) is the phase of *z*(*t*).

Thresholding is applied to adjust these characteristics, reducing noise by setting limits on the amplitude and frequency values:
(9)$$z^{\prime }(t)=\left\{\begin{array}{ll} z(t), & \mathrm{if}A(t)<\mathrm{threshold}\mathrm{and}\omega \left(t\right)<\mathrm{threshold}\\ 0, & \mathrm{otherwise} \end{array}\right.$$

The processed signal is reconstructed using the inverse Hilbert transform, aiming to retain the original signal information while minimizing new distortions. The primary advantage of this method is preserving the signal’s information while analyzing its instantaneous amplitude and frequency. However, the Hilbert transform struggles with complex signals containing multiple frequency components, limiting its effectiveness in broadband signals.
(3)CEEMDAN Model

The complete ensemble empirical mode decomposition with adaptive noise (CEEMDAN) model is an advanced signal processing method introduced by Dragomiretskiy et al. [16]. This method enhances the original empirical mode decomposition (EMD) by incorporating white noise to improve the stability and precision of the decomposition results. CEEMDAN is particularly effective in handling non-stationary and non-linear signals through a process involving signal decomposition, thresholding, and reconstruction. By incorporating white noise, CEEMDAN enhances the stability and precision of empirical mode decomposition (EMD), producing consistent results across multiple decompositions.

Initially, CEEMDAN decomposes a given signal *x*(*t*) using a series of white noise sequences *w_i_*(*t*). The decomposition is expressed as:
(10)$$x(t)=\sum_{j=1}^{m}I_{j}^{i}(t)+r_{k}(t)$$
where ${\mathrm{IMF}}_{j}^{i}(t)$ is the *j*-th intrinsic mode function of the *i*-th white noise series, and $r_{k}(t)$ is the residual. Averaging the results from multiple white noise series yields stable IMFs:



(11)
$${\mathrm{IMF}}_{j}(t)=\frac{1}{n}\sum_{i=1}^{n}I_{j}^{i}(t)$$



These IMFs represent distinct frequency components, partitioning the signal’s energy across the time and frequency domains.

Next, the decomposed IMFs undergo thresholding to eliminate noise. Techniques such as soft and hard thresholding are applied. For each IMF component, soft thresholding is expressed as:
(12)$${\mathrm{IMF}}_{j^{\prime }}(t)=\mathrm{sgn}{(\mathrm{IMF}}_{j}(t))\cdot \max({|\mathrm{IMF}}_{j}(t)|-\lambda_{j},0)$$

Hard thresholding is represented as:
(13)$${\mathrm{IMF}}_{j^{\prime }}(t)=\left\{\begin{array}{ll} {\mathrm{IMF}}_{j}(t), & \mathrm{if}{|\mathrm{IMF}}_{j}(t)|>\lambda_{j}\\ 0, & \mathrm{otherwise} \end{array}\right.$$

The threshold $\lambda_{j}$ is chosen based on the signal’s characteristics and noise level, often determined through experimentation and tuning. Adaptive thresholding can further optimize noise reduction.

Finally, the processed IMFs are summed to reconstruct the denoised signal:
(14)$$x^{\prime }(t)=\sum_{j=1}^{m}I_{j^{\prime }}(t)+r(t)$$

### 2.2. Unsupervised Skip Autoencoder Model (USAM)

In this paper, after thoroughly analyzing and comparing various denoising techniques, we introduce an unsupervised skip autoencoder network model designed for signal denoising. USAM is a distinct class of neural network architectures that do not rely on labeled training data. Instead, they employ self-learning mechanisms to compress and reconstruct data by capturing the high-dimensional features of the input. This paradigm is particularly adept at processing signals affected by substantial noise. This model, employing internal encoder and decoder mechanisms, learns to reconstruct a clear signal from noisy inputs. At the same time, its multi-layered design ensures the capture of subtle nuances that are present in complex signals.

During the training phase, we employ deep image prior (DIP) and the U-Net structure [33] to enhance the spatial correlation of proper signals within the network. In the encoding and decoding processes, skip connection blocks are integrated based on the characteristics of the AE data to prevent the leakage of valuable signals and thus maintain an optimal performance in preserving them. Additionally, these skip connections enhance further feature extraction. The feature extraction blocks, composed of fully connected layers, normalization layers, activation function layers, and dropout layers, are principally engaged in extracting data characteristics, compressing noisy AE signal data, and reconstructing denoised AE data.

The encoding process in the fully connected layer can be described as follows:(15)$$a_{ei}=W_{ei}P_{i}+b_{ei}$$
where *W_ei_* and *b_ei_*, respectively, represent the weight matrix and bias of the *i*-th layer’s fully connected layer during the encoding process. *a_ei_* denotes the output of the *i*-th fully connected layer, while *P_i_* signifies the input data.

Following the fully connected layer, a normalization layer is placed as the second layer in each feature extraction block to prevent the internal covariate shift. Throughout the iterative processes within the network, updating the parameters in lower layers can alter the distribution of the data in higher layers. Such disparities between the input and output data distributions can lead to cumulative errors, reducing the network’s fitting accuracy. By incorporating a normalization layer, training speeds can be enhanced, and the phenomenon of gradient vanishing can be mitigated.

The output from the normalization layer is then conveyed to a ReLU non-linear activation function layer, which can be expressed as follows:(16)ReLUx=0, x<0x, x≥0=max0,x

Following the activation layer, a dropout layer is introduced. The dropout layer is a widely utilized regularization technique aimed at preventing or mitigating the issue of model overfitting by randomly turning off a subset of neurons during the training phase. This method helps to enhance the generalization of the model to new data.

After each encoding block abstracts features from the AE data, multiple decoding blocks are employed to reconstruct the AE data. The architecture of the decoder mirrors that of the encoder, and the final output of each decoder block can be expressed as follows:(17)$${\hat{O}}_{di}=ReLU(W_{di}Q_{i}+b_{di})$$

*W_di_* and *b_di_* denote the weight matrix and bias for the *i*-th layer in the decoding process. *Q_i_* represents the input data for the *i*-th decoder block.

Subsequently, the feature extraction blocks are coupled with skip connections to form a symmetric end-to-end network model, which concludes with a fully connected layer and a linear activation function layer, outputting a reconstructed waveform signal that matches the dimensional attributes of the original AE waveform signal.

The network architecture is depicted in Figure 1. The model, principally comprising encoder, decoder, and skip connection (SK) blocks, is designed symmetrically. Each encoder and decoder block incorporates eight hidden layers, which include two fully connected layers, two normalization layers, two non-linear activation function layers, and two dropout layers. During the encoding phase, the model employs five encoder blocks to extract waveform features from the noise data, with the layer neurons in the fully connected layers decrementing sequentially from 64 to 32 to 16 to 8, and finally stabilizing at 8. In the decoding phase, four decoder blocks are utilized to reconstruct the denoised data, with the neurons in the fully connected layers of the decoders increasing in reverse order from 8 to 16 to 32 to 64. Additionally, the number of neurons in each skip connection block corresponds to the neuron count in the encoder blocks at the same level.

To preserve optimal training outcomes and prevent overfitting, an optimization strategy employing an early stopping mechanism is adopted. Training ceases automatically, and the best model parameters are saved when the loss on the validation set does not improve for ten consecutive epochs. The optimizer used is the Adam optimizer, enhancing the training efficiency and effectiveness of the model.

## 3. Evaluation Index

### 3.1. Simulated Signal

Due to the inherent noise in AE signals collected during experiments, obtaining a pure waveform signal is nearly impossible. To evaluate the performance of different denoising methods in a controlled environment and provide a consistent benchmark for comparison, employing simulated signals serves as an effective solution. In 1985, Mitrakovic et al. proposed a digital model to describe AE signals [34]. Simulated signals allow for the generation of clear signals without background noise, facilitating precise control over various signal parameters, such as the frequency, amplitude, and phase. Although the noise in practical AE signals is not broadband, introducing broadband noise into the simulated signals helps to test the generalization and adaptability of the denoising algorithms. The formula for the simulated signal is as follows:
(18)$$f\left(t\right)=\sum_{i=1}^{n}A_{i}e^{\left[{-Q}_{i}{{(t-t}_{i})}^{2}\right]}\sin[2\pi f_{i}\left(\begin{array}{c} {t-t}_{i}) \end{array}\right]$$

In the formula: *A_i_* represents the amplitude of the *i*-th superimposed signal; *Q_i_* denotes the attenuation factor of the *i*-th superimposed signal; *f_i_* is the delay time of the *i*-th superimposed signal; *n* signifies the number of superimposed signals within the simulated AE signal. The parameters selected for simulating the hydrating cement AE signal, as detailed in Table 1, are employed in this study.

To more realistically simulate the AE signals from concrete under cyclic loading, which include various noise components, a set of normally distributed white noise with a mean of 0 and a variance of 1 was incorporated into the simulated AE signals.

### 3.2. Fast Fourier Transform

The fast Fourier transform (FFT) is a formidable analytical tool for dissecting the frequency distribution of processed signals. It adeptly converts signals from the time domain into the frequency domain, thereby meticulously unveiling the various frequency components embedded within the signal. The transformation formula for the FFT is expressed as follows:(19)$$X(k)=\sum_{n=0}^{N-1}x(n)\cdot e^{-i2\pi kn/N}$$

*x*(*n*) represents the signal in the time domain, while *X*(*k*) denotes the components at different frequencies within the frequency domain. *N* signifies the total number of samples involved.

The FFT process is rapid, efficient, and exceptionally capable of revealing the dominant frequencies of a signal—those sections within the spectrum where the amplitude reaches its zenith. Through the application of FFT, the signal can discern the spectral shifts that occur before and after noise reduction, particularly noting how the energy redistributes among various frequencies and how the concentration areas shift. The identification of these dominant frequencies is crucial, as they typically harbor the most vital information within the signal.

Practical noise reduction algorithms aim to diminish the energy in non-dominant frequencies while preserving or enhancing the energy at dominant frequencies, thereby maintaining the signal’s core attributes and clarity. By juxtaposing the FFT results from before and after the noise reduction interventions, one gains a direct insight into the effectiveness of the noise reduction strategies, observing enhanced amplitudes at dominant frequencies and the effective suppression of non-essential frequency components.

Such assessments are invaluable as they facilitate a deeper understanding of the noise reduction algorithm’s performance, offering opportunities for fine-tuning and optimization to achieve more robust noise control and enhanced signal clarity. Thus, FFT transcends its foundational role as an analytical tool, becoming indispensable in complex signal-processing tasks, such as noise reduction. It provides a critical means of evaluation and optimization, ensuring both the high quality of the signal processing outcomes and their practical utility in diverse applications.

### 3.3. Signal-to-Noise Ratio

To assess the denoising capabilities of the model, we employed the signal-to-noise ratio (SNR) as a metric to evaluate the effects of noise reduction. The SNR is defined as the ratio of the power of the processed signal to the power of the background noise, quantifying the relative improvement in signal clarity and noise levels during the denoising process. By enhancing the SNR, one can directly observe how the denoising model effectively amplifies the crucial components of the signal while suppressing noise, thereby appraising its performance. We typically use decibels (dBs) as the unit for SNR, enabling a quantifiable comparison of the SNR values before and after processing, which illustrates the model’s noise-reduction efficacy under varying noise conditions. The formula for calculating the SNR is:(20)$$\mathrm{SNR}=10\times \mathrm{lg}[\sum_{\varphi =1}^{d}y_{\varphi }{\left(t\right)}^{2}/\sum_{\varphi =1}^{d}{\left[y_{\varphi }(t)-y_{\varphi }^{*}(t)\right]}^{2}]$$
where, $y_{\varphi }(t)$ and $y_{\varphi }^{*}(t)$ denote the original data and the data post denoising, respectively.

## 4. Analysis of Denoising Outcomes Based on Simulated Signals

### 4.1. Simulated Signals

The noise characteristics of the simulated signals differ from those of the actual AE signals. Noise in the real AE signals is more diverse and closely related to factors such as material properties and environmental conditions. However, broadband noise testing in simulated signals can provide an initial assessment of the adaptability of denoising algorithms in various noise environments.

The spectral characteristics of both the pure and noisy simulated AE signals were analyzed using FFT to compare their frequency components. Figure 2 presents the time-domain waveforms and spectral diagrams of the simulated AE signal and the noisy AE signal. As shown in Figure 2c, both the noise-free and noise-containing AE signals exhibit a “double-peak” phenomenon in their spectral graphs. However, the principal frequency of the pure AE signal is significantly lower than that of the noise-inclusive signal, with notable amplitude “oscillations” occurring at higher frequency ranges in the noisy signal. Additionally, the waveforms of the noisy AE signals are considerably more chaotic compared with their pure counterparts, demonstrating the adverse impact of noise on extracting critical parameters, such as ring-down counts, rise time, and the energy from AE signals. This interference complicates the accurate analysis of AE signals, making it more difficult to assess the expansion and characteristics of microcracks in concrete, and posing significant challenges for subsequent evaluations.

### 4.2. Analysis of Denoising

#### 4.2.1. Analysis of Wavelet Packet Denoising

Through the empirical selection of wavelet packet base functions for the AE signals, the Daubechies3 wavelet function (‘db3’) was chosen as the primary wavelet packet function due to its compact support, orthogonality, regularity, and near symmetry. Using the ‘db3’ wavelet packet, the AE signals are decomposed into four levels. Noise reduction is then achieved through an adaptive method that optimally decomposes the wavelet packet levels and applies an adaptive hard thresholding function, effectively filtering out noise while preserving the essential signal characteristics.

Figure 3a presents a comparison between the pure signal and its denoised counterpart, while Figure 3b contrasts the noisy signal with the denoised version. Figure 3c shows the spectral graphs of the pure signal after applying FFT, and Figure 3d compares the spectral graphs of the pure and denoised signals after FFT. The figures reveal that the wavelet packet decomposition method has not effectively eliminated noise. As shown in Figure 3a, the difference between the original pure signal and the denoised signal is evident. Ideally, the denoised signal should closely match the pure signal, preserving its essential characteristics and details. However, the results indicate that the denoised signal still contains significant noise components, as shown by the pronounced fluctuations in the waveform. This suggests that the wavelet packet decomposition method was ineffective in fully removing noise, and the denoised signal does not fully restore the purity and clarity of the original signal.

Additionally, as shown in Figure 3c,d, the amplitude of the denoised signal at specific frequencies has not significantly decreased; in some instances, the peaks are comparable to or even exceed those of the original signal. This indicates that the denoising process was ineffective in sufficiently removing noise and may have inadvertently weakened valuable signal components or introduced new interferences. These findings highlight the limitations of the wavelet packet decomposition method in this case, as it struggles to restore the pure signal’s structure and details effectively.

As shown in Figure 3b, the comparison between the noisy signal and its denoised counterpart further emphasizes the persistence of high-frequency noise and clutter, which hampers accurate signal recognition and analysis. Ideally, wavelet packet denoising should significantly reduce noise interference, resulting in a more uniform and smoother waveform. However, the visualization reveals that the denoised signal still contains substantial noise, falling short of the desired clarity. This confirms the suboptimal performance of the current denoising approach, demonstrating that it does not sufficiently enhance the signal quality.

#### 4.2.2. Analysis of Hilbert Transform Denoising

The subsequent diagram illustrates the impact of the Hilbert transform method (HTM) denoising treatment, featuring two key comparative visualizations. As shown in Figure 4a, the comparison between the pure signal and the denoised signal is presented, while Figure 4b contrasts the noisy signal with its denoised counterpart. After applying the Hilbert transform for denoising, it is clear that low-frequency noise components have been effectively reduced, leading to significant noise reduction in specific frequency bands and a notable enhancement in the signal’s overall smoothness. However, it is important to note that the Hilbert transform did not completely eliminate the high-frequency noise components. This residual high-frequency noise introduces minor fluctuations and distortions in the finer details of the signal. While the improvements in noise reduction are evident, the remaining noise still slightly impacts the signal’s overall integrity.

While the HTM offers clear advantages in noise reduction, particularly in eliminating low-frequency noise, its limitations in filtering high-frequency noise leave residual noise components in the high-frequency regions. This residual noise can negatively impact the overall analysis and application of the signal, reducing the effectiveness of the denoising process.

The expected smoothness and continuity are absent in the spectral graphs, as shown in Figure 4d. Instead, they reveal discontinuities and abnormal protrusions, indicating that the Hilbert transform did not sufficiently separate the signal from the noise. This inadequacy results in unnatural amplitude variations at specific frequency points, underscoring a critical limitation of the Hilbert transform as a denoising tool, particularly in applications where precise boundary recognition and a comprehensive smoothing of the frequency spectrum are crucial.

#### 4.2.3. Analysis of CEEMDAN Denoising

As shown in Figure 5, after undergoing CEEMDAN denoising, a simulated AE signal is decomposed into several intrinsic mode functions (IMFs), consisting of nine significant IMF components. A detailed analysis of these components reveals that each IMF exhibits distinct frequency characteristics and temporal variations, highlighting the complexity of the original signal. The characteristics of each IMF can be observed through both time-domain representations and spectral graphs, providing deeper insights into the signal’s intricate structure.

According to the graphical results, the first three IMFs are dominated by high-frequency noise. This noise is marked by irregular frequency patterns and large amplitude fluctuations, often caused by external environmental factors or equipment vibrations. These initial IMFs contribute significantly to noise interference, complicating the signal processing and analysis that follows. Therefore, they are identified as the primary sources of noise in the denoising process.

To remove these disturbances while retaining the essential signal content, the CEEMDAN method filters out the first three IMFs. In contrast, IMFs 4 through 9 exhibit lower frequencies and more stable fluctuations, suggesting that they capture the meaningful information from the original signal, rather than noise. The examination of these lower-frequency IMFs shows that they accurately represent the true physical phenomena and dynamics of the signal. As a result, IMFs 4 to 9 are selected as the foundation for signal reconstruction.

By combining IMFs 4 through 9, a denoised AE signal is reconstituted. This method preserves essential information from the original signal while greatly reducing the noise interference, providing a clearer and more accurate basis for further analysis and processing. This demonstrates the effectiveness of the CEEMDAN approach in improving the clarity and reliability of both signal decomposition and reconstruction.

Figure 6a provides a visual comparison between the original signal and the signal processed using CEEMDAN denoising, while Figure 6b contrasts the noisy signal with its denoised version. The figures demonstrate that, although the CEEMDAN method effectively reduces most of the high-frequency noise, it shows limitations in addressing low-frequency noise. In Figure 6a, the method successfully eliminates much of the high-frequency interference, resulting in a smoother and clearer signal. However, some low-frequency fluctuations persist, leaving the waveform less refined than desired, preventing the signal from reaching an ideal level of purity.

In Figure 6b, the CEEMDAN denoising method effectively suppresses high-frequency noise, significantly improving the waveform in those regions. However, some fluctuations remain in the low-frequency areas, indicating that the noise removal is not entirely complete. This highlights the method’s strength in reducing high-frequency noise but also points to the need for further refinement in addressing low-frequency noise.

In the spectral graphs of Figure 6c,d, the post-denoising reduction in amplitude across certain frequency areas indicates that CEEMDAN has effectively removed some of the noise components. However, the insufficient reduction at specific frequency points suggests that not all the noise has been fully eliminated. Additionally, discontinuities or abnormal peaks remain, reflecting possible over-suppression that could distort the signal or cause the loss of important information. This highlights a key area for improvement in the CEEMDAN method, specifically in balancing noise reduction with signal integrity across the full frequency spectrum.

#### 4.2.4. Analysis of USAM

Using the mean squared error (MSE) loss function during network training, Figure 7 demonstrates the effectiveness of noise reduction achieved through the USAM method, with two key comparative diagrams. Figure 7a compares the pure signal with the denoised signal after processing, and Figure 7b contrasts the noisy signal with its denoised version. A detailed analysis of these diagrams clearly highlights the USAM’s excellent performance in effectively eliminating noise from the signals.

In Figure 7a, the pure signal, representing an ideal state free from noise interference, is shown alongside the denoised signal. The pure signal’s waveform is smooth with well-defined details. After denoising through the USAM method, the restored signal closely resembles this purity. The waveform of the denoised signal appears smoother, with the noise disturbances effectively reduced. Notably, the process maintains the signal’s key features and details, indicating that the USAM method has strong noise reduction capabilities while preserving the essential information within the signal.

Figure 7b contrasts a noisy signal, filled with high- and low-frequency noise components that complicate the waveform and obscure valuable information, with the signal after undergoing denoising treatment. Through the application of USAM, most noise components in the noisy signal are effectively identified and filtered out. The denoised waveform appears significantly clearer and smoother, with reduced interference, greatly improving the signal’s readability and accuracy.

In the spectral graphs of Figure 7c,d, the amplitude at several frequency points in the denoised signal is significantly lower compared with the pure signal, suggesting that the denoising algorithm successfully identified and reduced the noise components. While the amplitude near the signal’s primary frequency components is also reduced, the useful information remains intact, demonstrating that the noise reduction process effectively removes noise without compromising the core content of the signal. Additionally, the denoised signal appears smoother, with no abnormal spikes or discontinuities, further indicating that the algorithm preserves the signal’s continuity and integrity during processing.

After the denoising process, a significant reduction in the signal’s amplitude is observed, with the denoised signal reaching 73.64% of the amplitude of the pure signal. This reduction highlights the considerable effect that noise removal can have on the amplitude characteristics of the signal, reflecting the efficiency of the denoising process in filtering out noise while still preserving the essential signal features.

During the denoising process, the conventional mean squared error (MSE) loss function may be inadequate due to the presence of outliers in the real-world signal data. MSE tends to disproportionately amplify the effect of outliers by squaring large errors, allowing them to exert an outsized influence on the overall training process. This can undermine the robustness of the model, as these outliers can skew the optimization process and negatively affect the model’s final performance, leading to less reliable denoising results.

To overcome this limitation, we propose using the Welsch loss function, a robust alternative designed to improve the model’s handling of outliers. The Welsch loss, widely employed in statistical applications to manage anomalies, offers superior suppression of outliers compared with the MSE loss function. This feature minimizes the influence of outliers on the training process, thereby enhancing the model’s resistance to noise and improving the overall performance in denoising tasks.

The Welsch loss function measures the difference between the predicted and true values using a non-linear function, with a hyperparameter that controls the sensitivity of the loss function to errors. The formula for the Welsch loss function is as follows:(21)$$\mathrm{WelschLoss}(y_{\mathrm{pred}},y_{\mathrm{true}})=\frac{1}{N}\sum_{i=1}^{N}\left(1-\exp\left(-0.5{\left(\frac{y_{\mathrm{true},i}-y_{\mathrm{pred},i}}{\delta }\right)}^{2}\right)\right)$$
where *N* represents the number of samples, while *y*_ture,*i*_ and *y*_pred,*i*_ correspond to the true value and the predicted value of the *i*-th sample, the hyperparameter δ, set at a value of 1.3, respectively.

As illustrated in Figure 8, the frequency amplitude graph highlights the results of the denoising process using the Welsch loss function during training. Compared with earlier methods, the amplitude of the denoised signal has increased to approximately 79.43% of the original signal’s amplitude. In the frequency domain, the denoised signal (shown in orange) retains the key amplitude characteristics in the low-frequency regions while effectively suppressing noise in the high-frequency segments. This improvement significantly enhances the signal-to-noise ratio, resulting in a clearer and more accurate signal representation.

This result highlights the Welsch loss function’s robust handling of outliers during the signal denoising process, as well as its effectiveness in suppressing high-frequency noise. Its ability to preserve essential signal characteristics while reducing unwanted noise demonstrates the Welsch loss function’s exceptional utility in advanced signal processing applications. The improved clarity and integrity of the denoised signal emphasize the Welsch loss function’s superior performance in achieving a higher fidelity, making it a powerful tool for enhancing the quality of signal representation.

### 4.3. Analysis of the SNR Value

Before applying the noise reduction methods, the original signal had a signal-to-noise ratio (SNR) of 0.9873502 dB, indicating significant noise content and reduced signal clarity. The wavelet packet denoising model was initially employed, leading to a slight increase in the SNR to 1.22879 dB. Although this improvement suggests some level of noise reduction, the modest SNR increase highlights the wavelet packet technique’s limited ability to substantially enhance the overall signal quality.

Subsequently, we assessed the Hilbert transform noise reduction model. After processing, the SNR achieved a significant gain, reaching 8.33811 dB, a marked improvement from the baseline. Compared with the wavelet packet denoising method, the Hilbert transform exhibited a superior performance in filtering noise and preserving signal fidelity, leading to a substantial enhancement in the overall signal quality.

Further analysis includes the CEEMDAN denoising model, which raised the SNR to 10.26474 dB. This improvement underscores CEEMDAN’s advanced capability in noise reduction, significantly enhancing the signal quality. Compared with the Hilbert transform, CEEMDAN achieved a more substantial SNR gain, demonstrating its superiority in managing complex noise environments and providing a more effective solution for improving signal clarity.

Lastly, the analysis of the USAM denoising model revealed a signal-to-noise ratio (SNR) gain of 10.743391 dB. Although this SNR improvement appears minor compared with CEEMDAN, SNR is only one of many metrics used to assess noise reduction performance. To provide a more comprehensive evaluation, we also analyzed the waveforms and frequency spectra after FFT transformation, as shown in the figures. Specifically, Figure 6a displays the waveform before and after CEEMDAN noise reduction, while Figure 6d shows the corresponding frequency spectrum. In contrast, Figure 8a presents the waveform after applying the USAM noise reduction model, and Figure 8b shows the corresponding frequency spectrum after FFT transformation.

From these figures, it is evident that, although the USAM model produced only a moderate SNR improvement, its performance in both the time and frequency domains is superior. The waveform in Figure 8a is smoother, with significantly less residual noise, and the frequency spectrum in Figure 8b shows effective noise band reduction while retaining essential signal components. This indicates that the USAM model excels at distinguishing signals from noise, resulting in enhanced signal quality and interpretability.

Therefore, while the SNR gain may seem modest, the improvements in both the waveform and spectrum achieved by the USAM model are more significant and intuitive. This demonstrates the great potential of the USAM approach in noise suppression, as it is capable of finely analyzing and eliminating noise components, leading to a substantial improvement in the overall signal quality.

In summary, this comparative analysis highlights the performance of various denoising methods in improving the SNR from an initial 0.9873502 dB to varying degrees. The wavelet packet approach had a limited effect, the Hilbert transform performed well, CEEMDAN proved notably effective, and USAM demonstrated superior noise reduction capabilities. Thus, selecting the appropriate noise reduction model is essential for enhancing the signal quality, especially in complex noise environments, where deep learning technology, such as USAM, shows the greatest potential and promise.

## 5. Analysis of Denoising Outcomes Based on Real Signals

Although simulated signal testing provides an initial evaluation of the denoising methods, the complexity of noise in actual AE signals necessitates experimental validation using real AE signals in this study, to further assess the effectiveness of the denoising techniques.

### 5.1. Experiment Process


(1)Tensile test of SFRC


The tensile mechanical properties were assessed through a uniaxial tension test, with the experimental setup depicted in Figure 9b. The loading rate was controlled at 0.2 mm/min, and the dimensions of the tensile specimens are shown in Figure 9a. The AE signals were carefully monitored using the advanced Express II-Micro eight-channel AE system and managed via the AE win software. As illustrated in Figure 9c, eight sensors were strategically placed on the specimen. To ensure the minimal attenuation of the AE signals due to contact gaps, high vacuum grease was applied as an acoustic coupling agent to secure the sensors onto the beam surfaces.

To minimize the interference from extraneous noise during the experiment, the AE acquisition system was preconfigured with an AE signal threshold of 35 dB, a peak definition time (PDT) of 50 µs, a hit definition time (HDT) of 300 µs, and a hit lockout time (HLT) of 1000 µs. As illustrated in Figure 9, part (a) provides a schematic representation of the experimental procedure, while part (b) shows the sensor arrangement. To ensure the comprehensive and accurate capture of the AE signals, two sensors were placed on the lateral surface of the specimen, with an additional two sensors positioned at both the upper and lower ends of the rear surface, resulting in a total of eight sensors.
(2)Tensile test of steel

The experimental material is used Q335 steel, based on the GB/T 228.1-2010 [35] standards and ASTM E8/E8M-11 [36] specifications. The structural dimensions of the tensile specimen and the arrangement of the acoustic emission sensors are shown in Figure 10. The specimen is a standard tensile specimen made of Q335 pipeline steel.

As illustrated in Figure 11, an MTS fatigue testing machine was employed for the experiment. The universal testing machine operated under displacement-controlled loading at a rate of 1 mm/min. The acoustic emission equipment parameters were consistent with those used in the previous section for the tensile testing of SFRC, ensuring uniformity in signal monitoring and analysis across both experimental setups.
(3)Flexural experiment of the reinforced concrete beams

The dimensions of all the tested RC beams are 150 mm (depth) × 100 mm (width) × 550 mm (span length), as shown in Figure 12. The concrete in this study was made up of ordinary Portland cement (PO42.5), the coarse aggregate (mechanically broken basalt gravel), and the fine aggregate, and the W/C is 0.33. Table 2 provides the labels and critical parameters of the tested RC beams.

The experiment was carried out using a hydraulic servo actuator testing machine equipped with a data acquisition system and an AE monitoring system, as depicted in Figure 13. All the RC beams were loaded vertically at a slow rate of 0.1 mm/min to ensure the accurate acquisition of the AE data. The AE signals were monitored using the Express II-Micro eight-channel AE system, managed through the AE win software. A total of eight sensors were strategically arranged on the specimen, as shown in Figure 1, with high vacuum grease applied as an acoustic coupling agent to secure the sensors to the beam surfaces and minimize AE signal attenuation due to contact gaps.

### 5.2. Experimental Signal Noise Reduction

Before applying denoising procedures using models, it is essential to optimize the model’s convergence rate and enhance its performance while avoiding issues like numerical instability. To achieve this, the AE waveform signals collected during the experimental processes were divided into the training and validation datasets, with 80% allocated for training and 20% for validation. This division ensures that the model is effectively trained while preserving a dataset for the performance evaluation.

In the tensile test of SFRC, Figure 14 illustrates the loss rate trajectories for both the training and validation datasets during the model training phase. The loss curves for the three specimens follow similar patterns, indicating consistent behavior across different datasets. This uniformity in the loss trends suggests that the model performs robustly, generalizing effectively across varied samples. The model captures the underlying dynamics with precision, ensuring stability and reliability across the evaluated scenarios.

As shown in Figure 14a, after ten epochs, the loss rate for the training set in specimen K1 stabilized at approximately 0.021, while the validation set’s loss rate decreased to approximately 0.022. The training process was halted at 40 epochs due to the early stopping strategy, with final loss rates of 0.0188 for the training set and 0.0198 for the validation set.

Similarly, Figure 14b shows that for specimen K2, the training set’s loss rate stabilized at 0.018, while the validation set’s loss rate dropped to 0.024 after ten epochs. The training was halted at 50 epochs, resulting in final loss rates of 0.019 for the training set and 0.014 for the validation set.

Figure 14c illustrates the performance of specimen K3, where the training set’s loss rate stabilized at 0.02, and the validation set’s loss rate decreased to 0.016 after ten epochs. The training process stopped at 44 epochs, with final loss rates of 0.017 for the training set and 0.014 for the validation set. This consistency across all three specimens demonstrates the model’s reliability in achieving a stable performance.

In the tensile test of steel, as depicted in Figure 15, the loss rates for both the training and validation sets were tracked across the three specimens. For specimen S1, Figure 15a shows that the loss rate for the training set stabilized at approximately 0.0091 after ten epochs, while the validation set’s loss rate decreased to approximately 0.0095. The training process was halted at 35 epochs, with final loss rates of 0.0088 for the training set and 0.0092 for the validation set.

Similarly, in Figure 15b, specimen S2 exhibited a comparable pattern, with the training set’s loss rate stabilizing at 0.0090, while the validation set’s loss rate decreased to 0.010 after ten epochs. The training concluded at 35 epochs, resulting in final loss rates of 0.009 for the training set and 0.0098 for the validation set.

Figure 15c illustrates the consistent behavior of specimen S3, where the training set’s loss rate stabilized at 0.0093, and the validation set’s loss rate decreased to approximately 0.010 after ten epochs. The training process was stopped at 33 epochs, yielding final loss rates of 0.0091 for the training set and 0.0097 for the validation set. The slightly higher loss rate in the validation set compared with the training set suggests that the model does not suffer from overfitting.

In the flexural experiment of reinforced concrete beams, as shown in Figure 16, the three specimens exhibited similar patterns in training and validation loss rates. In Figure 16a, specimen F1 shows that after ten epochs, the training set’s loss rate stabilized around 0.009, while the validation set’s loss rate decreased to approximately 0.0092. The training process was halted at 34 epochs, with final loss rates of 0.0087 for the training set and 0.0088 for the validation set.

Similarly, in Figure 16b, the loss rate for the training set in specimen F2 stabilized at 0.009 after ten epochs, while the validation set’s loss rate decreased to 0.0095. The training process for F2 was stopped at 34 epochs, yielding final loss rates of 0.0084 for the training set and 0.0086 for the validation set.

Finally, Figure 16c shows the results for specimen F3, where the training set’s loss rate stabilized at 0.009, and the validation set’s loss rate decreased to 0.0092 after ten epochs. The training was halted at 34 epochs, resulting in final loss rates of 0.0087 for the training set and 0.0088 for the validation set. The marginally higher loss rate observed in the validation set compared with the training set indicates that the model effectively avoids overfitting.

### 5.3. Experimental Waveform Analysis with FFT Dominant Frequency Results

Figure 17a illustrates the transformation of the AE signals before and after noise reduction during the tensile testing of SFRC specimens. In the original signal, various interferences and noise introduce significant fluctuations and irregular peaks, masking the true information within the signal. This distortion makes it challenging to accurately interpret the data, as the noise obscures the essential details. However, after applying noise reduction techniques, the AE signal’s waveform becomes noticeably smoother and more coherent. The extraneous noise is effectively reduced, resulting in a clearer representation of the signal’s key features, allowing for more a precise analysis and interpretation of the AE data.

Figure 17b–d present the spectral graphs of the AE signals. In Figure 17b, the spectrum of the original signal shows a wide distribution of frequencies, with the energy spread across multiple frequency bands. This indicates a significant presence of interfering noise, which dilutes the meaningful information in the signal and complicates its analysis.

In contrast, Figure 17c shows the spectrum after noise reduction, where the spectral graph is more concentrated, with the primary energy focused on dominant frequencies that carry the essential information. This focused energy concentration in specific frequency bands demonstrates the effectiveness of the noise reduction process, which filters out irrelevant noise frequencies while preserving the valuable components of the signal.

Figure 17d provides a comparative view of the spectral diagrams of both the original and noise-reduced signals. This comparison clearly highlights the improvements that are achieved through noise reduction. The noise-reduced signal exhibits a more refined and concentrated spectral distribution, with less energy dispersion across non-essential frequency bands. This optimized spectral energy distribution enhances the clarity and usability of the AE signal, making it more reliable for subsequent analysis and decision-making processes in civil engineering applications.

Figure 18a illustrates the time-domain waveform of the AE signals before and after noise reduction during the tensile testing of steel specimens. The original waveform shows significant noise and irregularities, which obscure the actual signal information. After noise reduction, the waveform becomes noticeably smoother and more coherent, with most of the extraneous noise effectively minimized, allowing the essential features of the signal to stand out more clearly and be more easily interpreted.

Figure 18b–d present the spectral analysis. Figure 18b displays the original signal’s broad energy distribution, indicating substantial noise interference across multiple frequency bands. In Figure 18c, after noise reduction, the energy becomes more concentrated, with dominant frequencies more clearly defined, demonstrating the effectiveness of the noise reduction process in filtering out irrelevant noise. Finally, Figure 18d compares the spectra before and after noise reduction, highlighting the refined spectral clarity. This improvement significantly enhances the reliability of the AE data analysis during the tensile testing of steel specimens, providing more accurate and interpretable results.

Figure 19a illustrates the time-domain waveform of the AE signals before and after noise reduction during the tensile testing of steel specimens. The original waveform shows significant noise and irregularities, which obscure the actual signal information. After noise reduction, the waveform becomes noticeably smoother and more coherent, with most of the extraneous noise effectively minimized, allowing the essential features of the signal to stand out more clearly and be more easily interpreted.

Figure 19b–d present the spectral analysis. Figure 19b displays the original signal’s broad energy distribution, indicating substantial noise interference across multiple frequency bands. In Figure 19c, after noise reduction, the energy becomes more concentrated, with dominant frequencies more clearly defined, demonstrating the effectiveness of the noise reduction process in filtering out the irrelevant noise. Finally, Figure 19d compares the spectra before and after noise reduction, highlighting the refined spectral clarity. This improvement significantly enhances the reliability of the AE data analysis during the tensile testing of steel specimens, providing more accurate and interpretable results.

## 6. Conclusions

This study successfully applied deep learning techniques to denoise AE signals, improving the signal quality and enabling more accurate assessments of the materials’ structural integrity. The experimental results confirm the superior performance of the proposed unsupervised skip autoencoder model (USAM) when compared with traditional noise reduction methods. The following conclusions can be drawn:(1)This study effectively evaluates the denoising performance of various methodologies on simulated AE signals, emphasizing each method’s ability to preserve the signal integrity and clarity. In particular, USAM, trained with the Welsch loss function, significantly improved the signal quality, increasing the signal-to-noise ratio (SNR) from 0.9873502 dB to approximately 10.743391 dB. This substantial enhancement demonstrates the potential of deep learning in noise reduction, enabling a detailed analysis and effective noise resolution, which greatly enhances the overall signal quality. Such advancements streamline detection processes and improve the reliability of structural health diagnostics, which are critical for ensuring safety and efficiency in engineering applications.(2)The applicability of these denoising methods in the actual AE data for structural health monitoring appears promising. Although wavelet packet denoising was adequate, its noise reduction capacity was somewhat limited. In contrast, denoising through Hilbert transformation exhibited a superior performance in filtering low-frequency noise components, and CEEMDAN denoising demonstrated considerable potential in managing non-stationary and non-linear signals. However, further refinement of the data, especially in the low-frequency regions, is requisite to achieve complete noise elimination. This nuanced enhancement is essential to optimize the precision and reliability of structural diagnostics, ensuring more accurate interpretations of the structural integrity and early warnings of potential failures.(3)The successful application of USAM in AE signals collected during tensile tests of SFRC, tensile tests of Q335 steel, and flexural tests of reinforced concrete (RC) beams underscores their practical relevance. The deep learning model’s robust noise reduction capabilities and ability to maintain essential signal features effectively eliminated the noise and enhanced the clarity and accuracy of AE signals after denoising in both the waveform and spectral graphs.

## Figures and Tables

**Figure 1 sensors-24-06145-f001:**
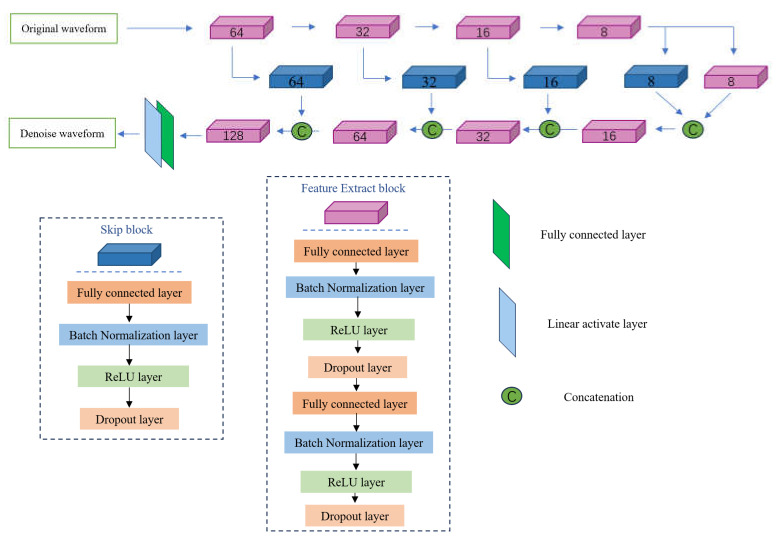
Model diagram.

**Figure 2 sensors-24-06145-f002:**
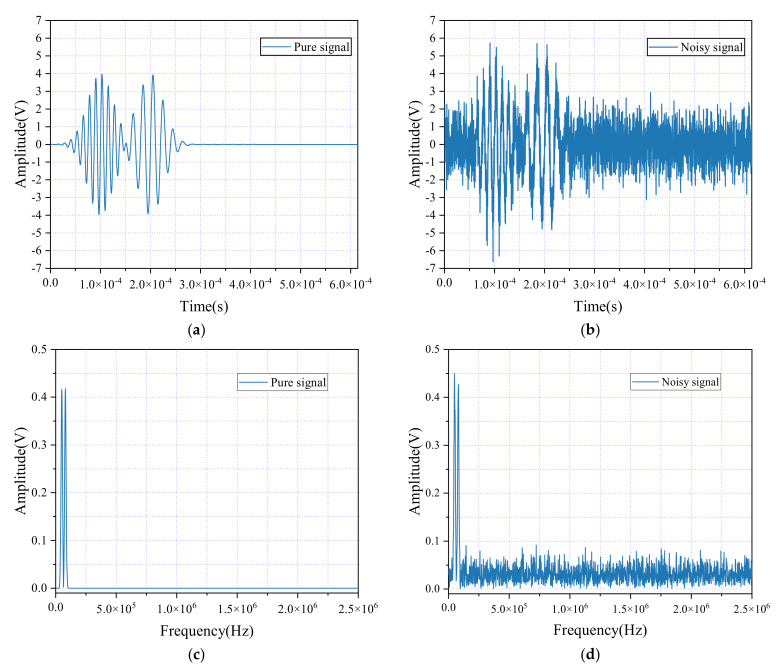
Time-domain waveforms and spectral diagrams of simulated AE signals and noisy AE signals. (**a**) Waveform diagram of the simulated AE signal; (**b**) Waveform diagram of the simulated AE signal with noisy; (**c**) Spectrum diagram of the simulated AE signal; (**d**) Spectrum diagram of the simulated AE signal with noisy.

**Figure 3 sensors-24-06145-f003:**
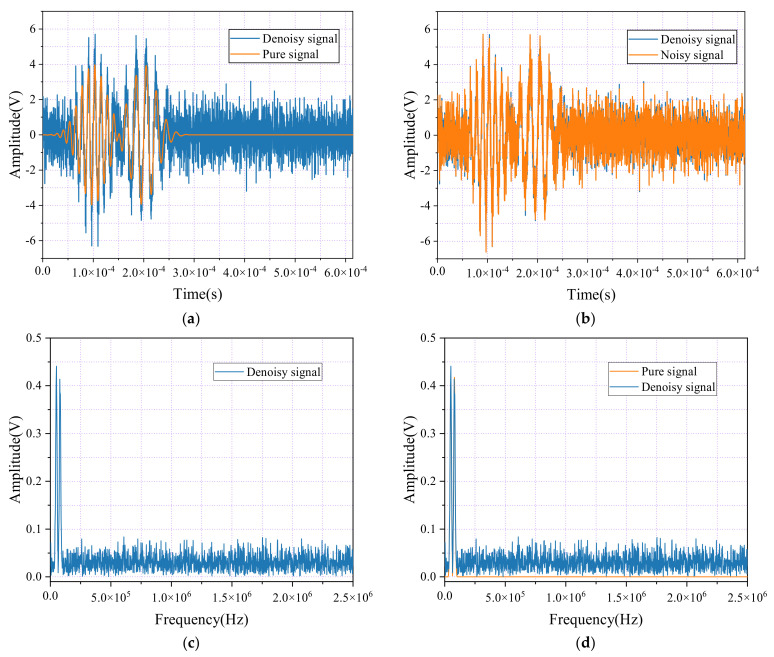
Time-domain waveforms and spectral diagrams of simulated AE signals, noisy AE signals, and denoise AE signal by the wavelet packet. (**a**) Waveforms of the AE signal and denoise AE signal; (**b**) Waveforms of the AE signal with noisy and denoise AE signal; (**c**) Spectral diagram of the AE signal and denoise AE signals. (**d**) Spectral diagram of the AE signal with noisy and denoise AE signals.

**Figure 4 sensors-24-06145-f004:**
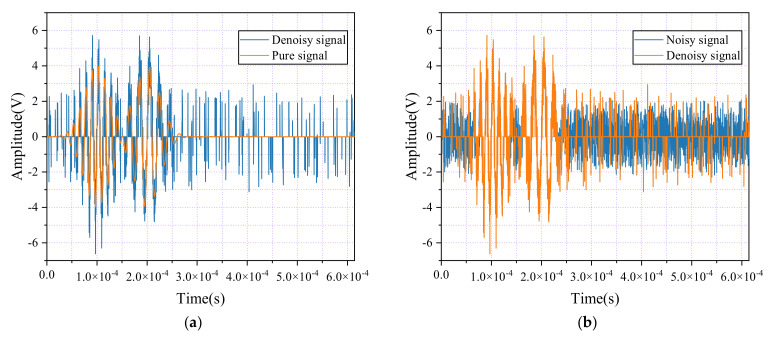
Time-domain waveforms and spectral diagrams of simulated AE signals, noisy AE signals, and denoise AE signals by HHT. (**a**) Waveforms of AE signal and denoise AE signal. (**b**) Waveforms of AE signal with noisy and denoise AE signal. (**c**) Spectral diagrams of AE signal and denoise AE signal. (**d**) Spectral diagrams of AE signal with noisy and denoise AE signal.

**Figure 5 sensors-24-06145-f005:**
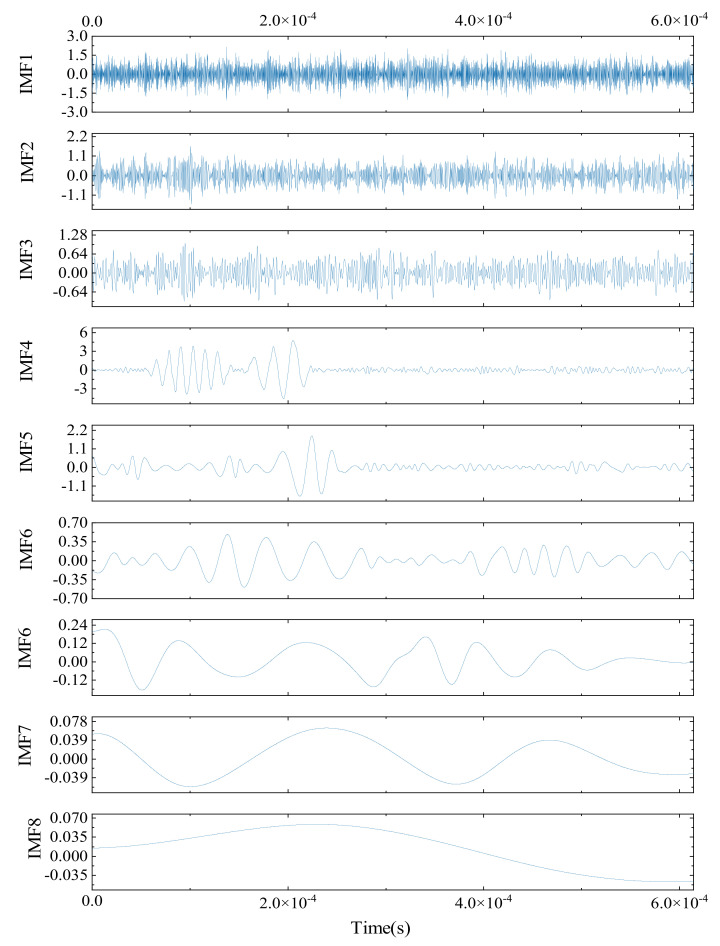
Diagram of CEEMDAN decomposition.

**Figure 6 sensors-24-06145-f006:**
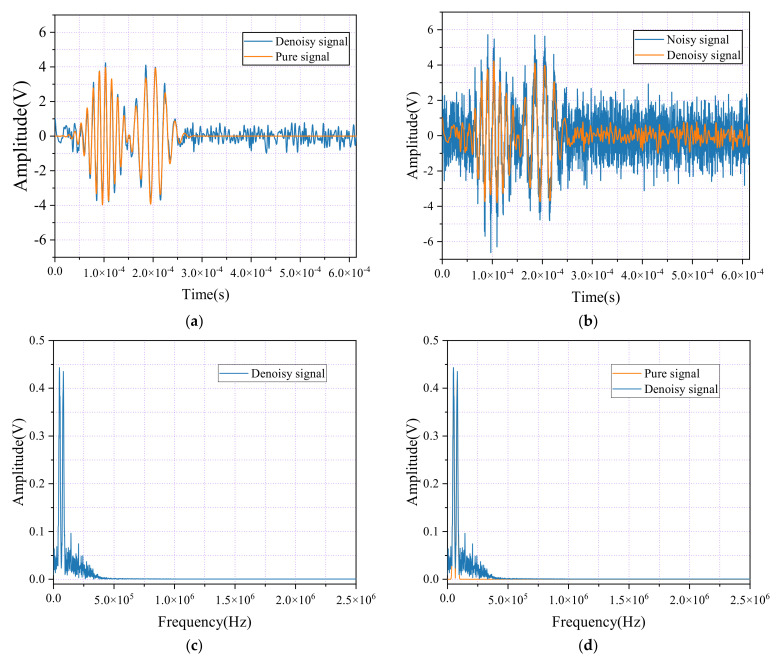
Time-domain waveforms and spectral diagrams of simulated AE signals, noisy AE signals, and denoise AE signals by CEEMDAN. (**a**) Waveforms of the AE signal and denoise AE signal; (**b**) Waveforms of the AE signal with noisy and the denoise AE signal; (**c**) Spectral diagrams of the AE signal and denoise AE signal; (**d**) Spectral diagrams of the AE signal with noisy and the denoise AE signal.

**Figure 7 sensors-24-06145-f007:**
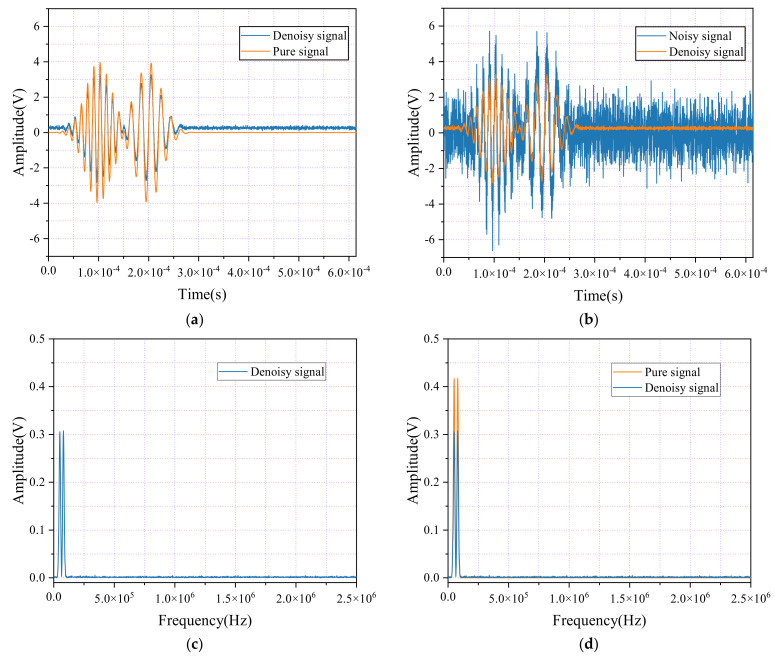
Time-domain waveforms and spectral diagrams of simulated AE signals, noisy AE signals, and denoise AE signals by USAM and MSE. (**a**) Waveforms of the AE signal and denoise AE signal; (**b**) Waveforms of the AE signal with noisy and the denoise AE signal; (**c**) Spectral diagrams of the AE signal and denoise AE signal; (**d**) Spectral diagrams of the AE signal with noisy and the denoise AE signal.

**Figure 8 sensors-24-06145-f008:**
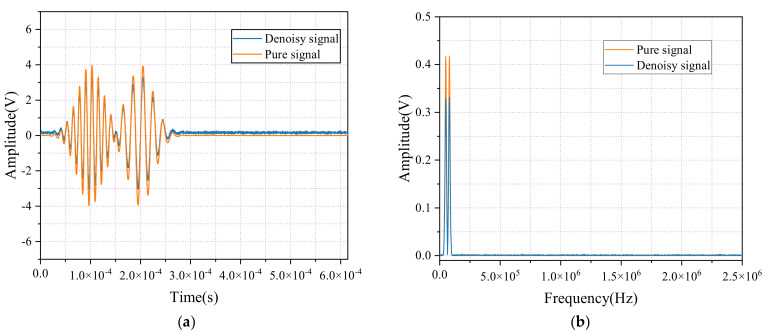
Time-domain waveforms and spectral diagrams of simulated AE signals and denoise AE signals by USAM and Welsch loss. (**a**) Waveforms of the AE signal and the denoise AE signal; (**b**) Spectral diagrams of the AE signal with noisy and the denoise AE signal.

**Figure 9 sensors-24-06145-f009:**
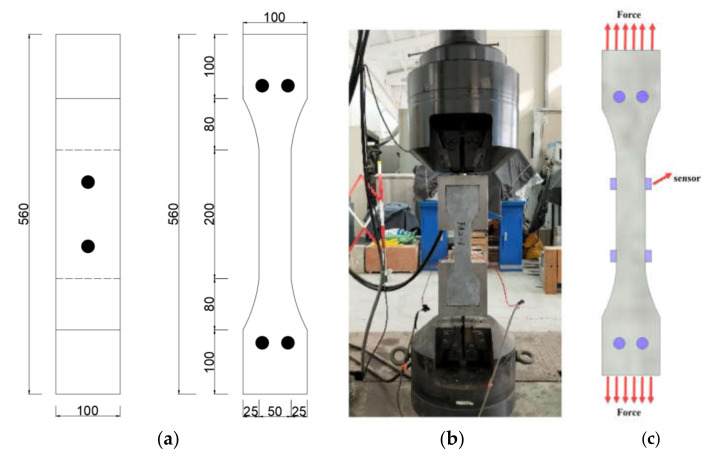
Experimental procedure and sensor layout. (**a**) Experimental procedure; (**b**) Experimental procedure; (**c**) Arrangement of the sensors.

**Figure 10 sensors-24-06145-f010:**
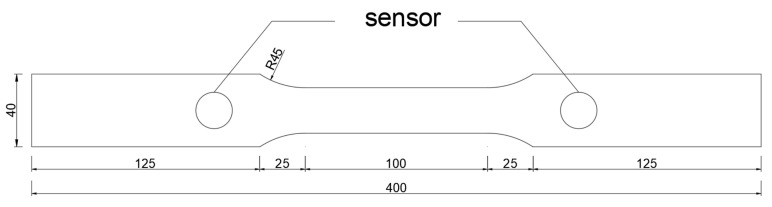
Diagram of the specimen dimensions and the arrangement of the acoustic emission sensors.

**Figure 11 sensors-24-06145-f011:**
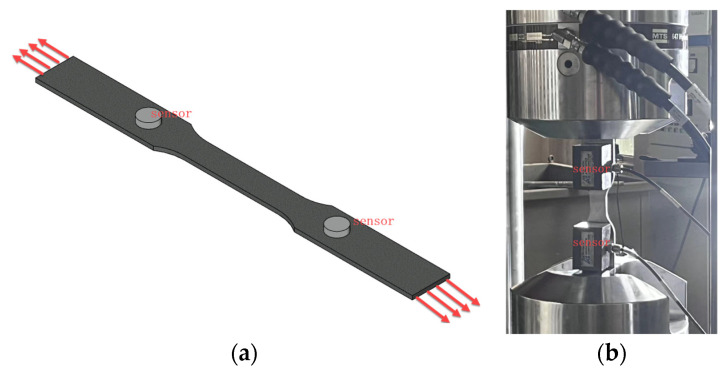
Tensile testing schematic and experimental procedure. (**a**) Tensile testing schematic; (**b**) Experimental procedure.

**Figure 12 sensors-24-06145-f012:**
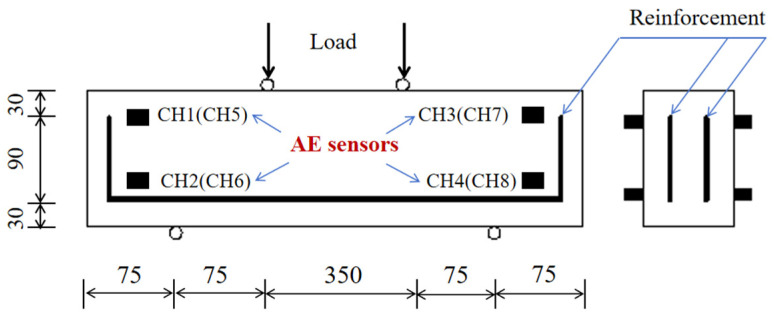
Analysis of SNR. The dimensions of the RC beam and the arrangement of the AE sensors.

**Figure 13 sensors-24-06145-f013:**
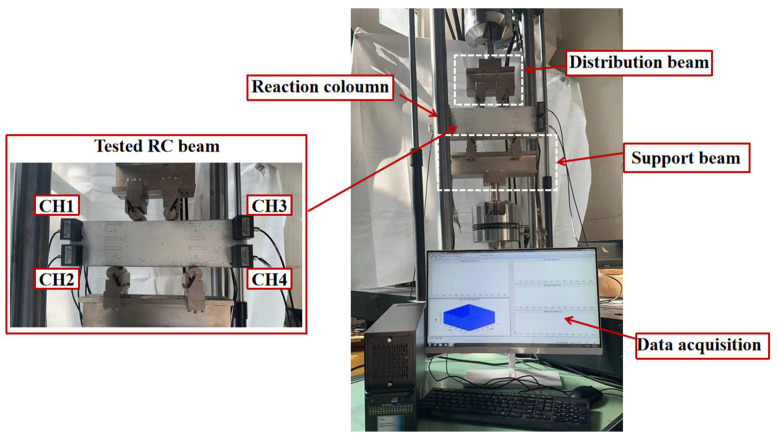
Test setup.

**Figure 14 sensors-24-06145-f014:**
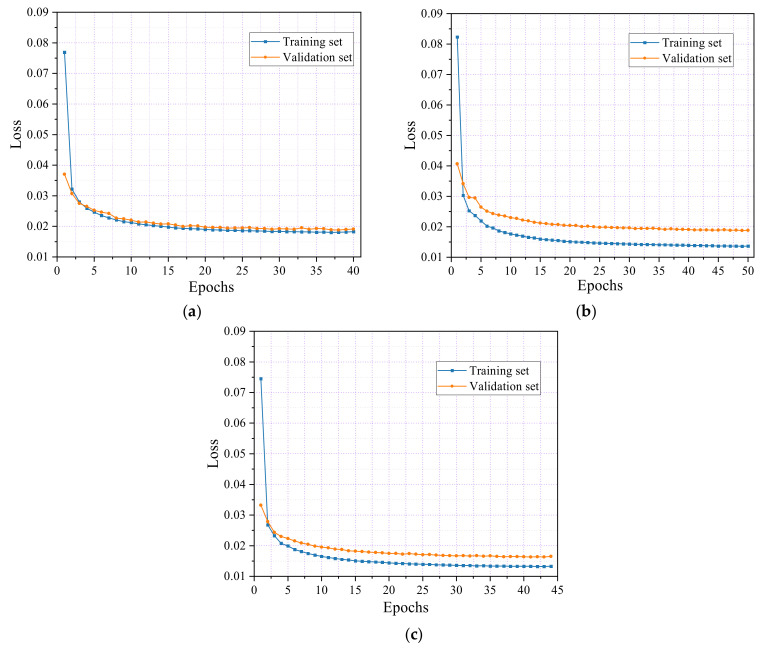
Training and validation loss curves of SFRC. (**a**) Specimen K1; (**b**) Specimen K2; (**c**) Specimen K3.

**Figure 15 sensors-24-06145-f015:**
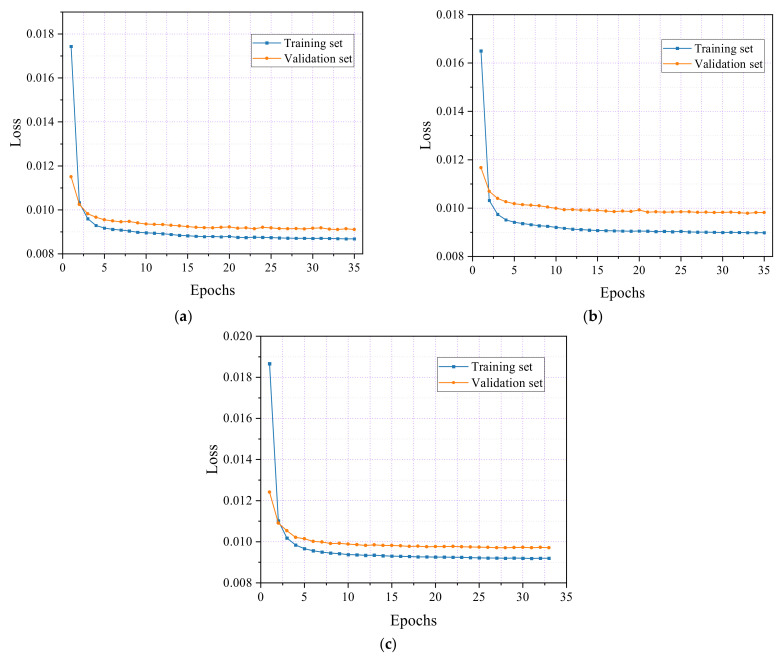
Training and validation loss curves of steel tensile. (**a**) Specimen S1; (**b**) Specimen S2; (**c**) Specimen S3.

**Figure 16 sensors-24-06145-f016:**
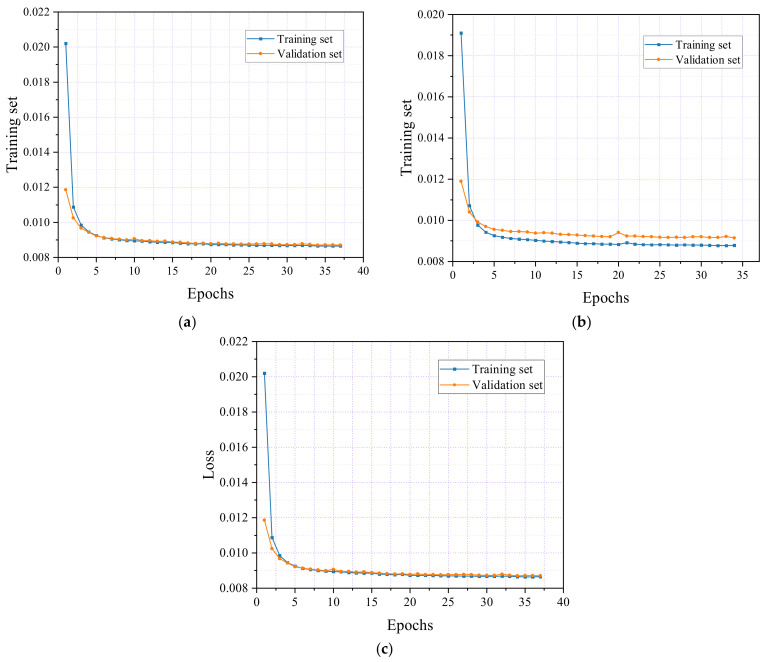
Training and validation loss curves of the reinforced concrete beam flexural. (**a**) Specimen F1; (**b**) Specimen F2; (**c**) Specimen F3.

**Figure 17 sensors-24-06145-f017:**
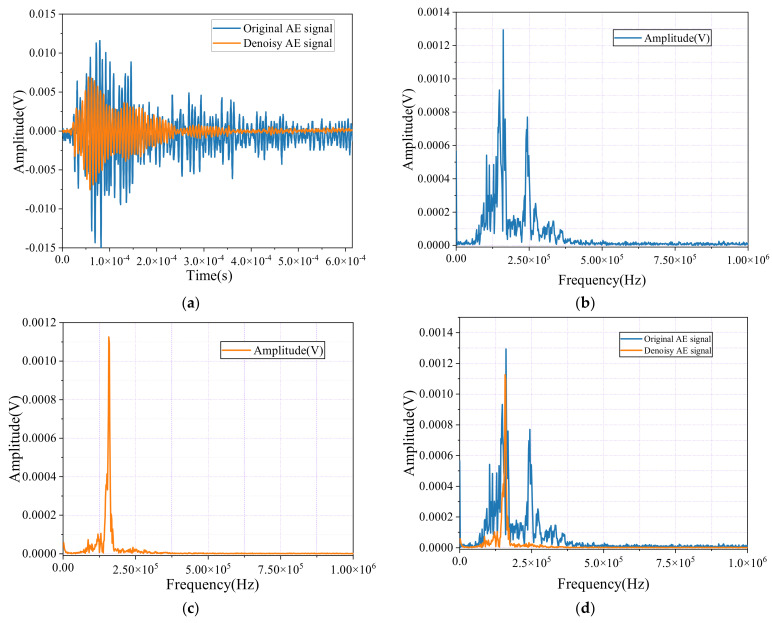
Time-domain waveforms and spectral diagrams of noisy AE signals and denoise AE signals of SFRC. (**a**) AE signal and denoise AE signal; (**b**) AE signal; (**c**) Denoise AE signal; (**d**) AE signal and denoise AE signal.

**Figure 18 sensors-24-06145-f018:**
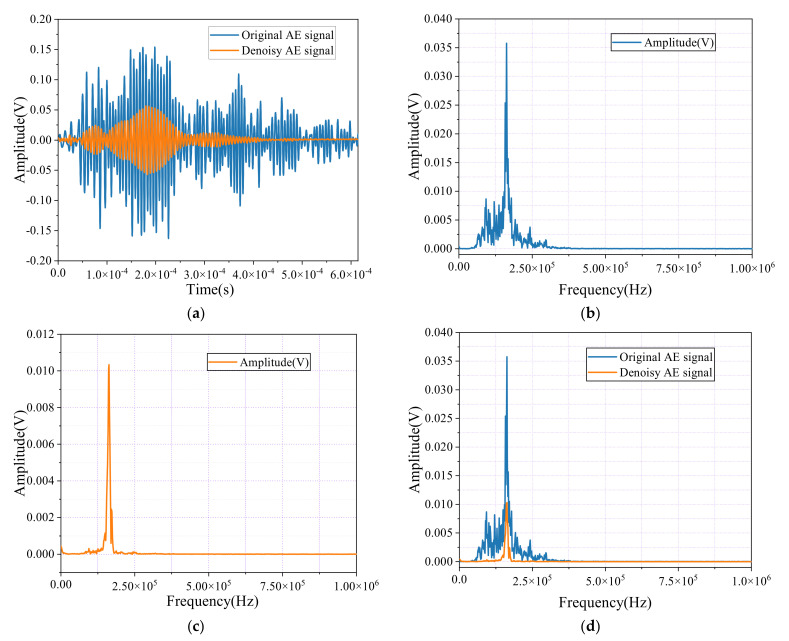
Time-domain waveforms and spectral diagrams of noisy AE signals and denoise AE signals. (**a**) AE signal and denoise AE signal; (**b**) AE signal; (**c**) Denoise AE signal; (**d**) AE signal and denoise AE signal.

**Figure 19 sensors-24-06145-f019:**
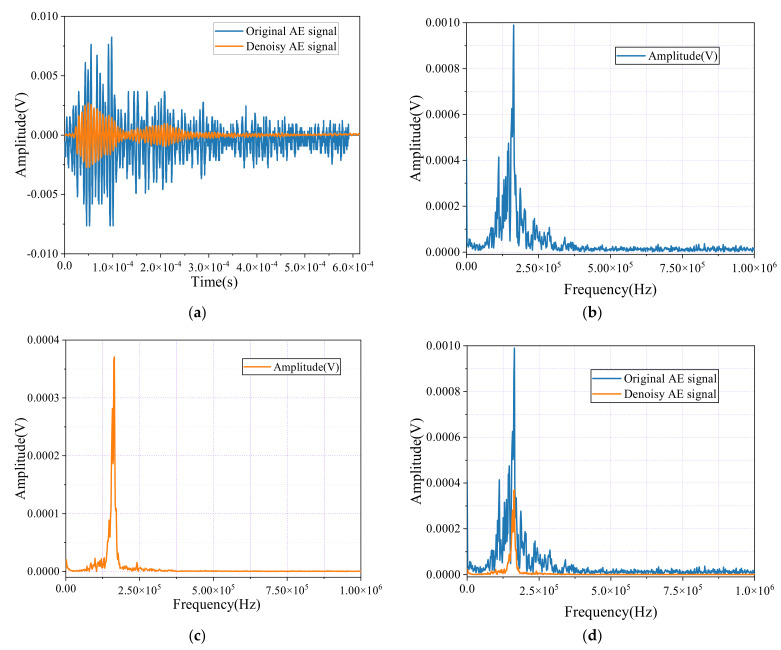
Time-domain waveforms and spectral diagrams of noisy AE signals and denoise AE signals. (**a**) AE signal and denoise AE signal; (**b**) AE signal; (**c**) Denoise AE signal; (**d**) AE signal and denoise AE signal.

**Table 1 sensors-24-06145-t001:** Parameter table of the analog AE signal.

Parameter	N	A_1_	A_2_	Q_1_	Q_2_	t_1_/s	t_2_ (s)	t(s)	f_1_/k	f_2_/kHz	Sampling Point
value	2	4	4	7.62 × 10^8^	7.62 × 10^8^	1 × 10^−4^	2 × 10^−4^	1 × 10^−3^	80	50	3072

**Table 2 sensors-24-06145-t002:** Mix ratio of RC specimens (kg/m^3^).

W/C	Water	Cement	Fine Aggregate	Coarse Aggregate	Coarse Aggregate Size (mm)	Steel Reinforcement Diameter (mm)	Specimen
0.33	230.78	720.55	676.67	1015.00	4.75-9.5	10/12/14	RC1-1/-2/-3

## Data Availability

The data that support the findings of this study are available from the corresponding author upon reasonable request.

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
