# Peer review of "Study of Acoustic Emission Signal Noise Attenuation Based on Unsupervised Skip Neural Network"

_sensors, 2024, doi:10.3390/s24186145_

Round 1
Reviewer 1 Report
Comments and Suggestions for Authors
(1) The paper addresses noise reduction strategies for acoustic emission testing, implementing a deep learning technique based on denoising implementations. After implementing a theoretical-based denoising approach, referring to an artificial signal and to multiple filtering techniques, the study applies the developed technique with regard to a few representative experimental tests, showing potential effectiveness.
(2) Even though the paper might contribute to the literature in terms of methodological guidance and technical insights, it cannot be considered to be suitable for publication in its current form and requires revisions. Please find attached a detailed report with the referee’s comments and recommendations.

Comments on the Quality of English LanguageEnglish is overall fine, but grammar and syntax checks are suggested.
Reviewer 2 Report
Comments and Suggestions for Authors
The paper discusses the use case of deep-learning based autoencoder model to denoize acoustic emission signals in concrete and steel structures. The authors claim that the deep-learning autoencoder model approach provides superior noise suppression compared to DSP approaches, and, as a result, significant improvement in SNR value is achieved.
The authors need to address the following comments:
Comments:
-
In lines 604-607, the authors mention that the CEEMDAN based statistical approach leads to SNR improvement of about 10dB. The deep-learning noise reduction model doesn’t significantly improve the SNR from what CEEMDAN attains. If the SNR improvement is minuscule, then what is the benefit of going towards a deep-learning based autoencoder model?
-
In line 691, the authors mention that the loss rate for the training set approaches 0.191, which doesn’t match the corresponding figure. Please address this discrepancy.
-
Image clarity in Figure-1 is poor. Please provide an enlarged and clear image. If it is needed, utilize the whole page.
-
Please redo Figures-3, 4, 6, 7, including legends such as “Denoisy signal” and all four figure captions.
-
Line 638 mentions Figure1(a) erroneously. Please correct it.
-
Line 669, the Figure# is incorrect. Please correct it.
-
Figure-14,15,16 need proper referencing to the individual figures, and the figure caption should have relevant details. Please improve the figures.
-
What differentiates Figure 19, from, say Figure 17, 18? The narration and explanations are very similar in each. Also, for each of these figures, please improve the tick-mark font size, make all the sub-figures of the same size, and rework the figure captions.
The authors need to improve the figure legend text, figure captions.
Round 2
Reviewer 2 Report
Comments and Suggestions for Authors
The authors have reasonably addressed all the comments.
Author Response
Thank you for your review.